# A Silica–Lignin Hybrid Filler in a Natural Rubber Foam Composite as a Green Oil Spill Absorbent

**DOI:** 10.3390/polym14142930

**Published:** 2022-07-20

**Authors:** Yati Mardiyati, Anna Niska Fauza, Onny Aulia Rachman, Steven Steven, Sigit Puji Santosa

**Affiliations:** 1Materials Science and Engineering Research Group, Faculty of Mechanical and Aerospace Engineering, Institut Teknologi Bandung, Jl. Ganesha 10, Bandung 40132, Indonesia; annaniska23@gmail.com (A.N.F.); onnyaulia7@gmail.com (O.A.R.); steven@material.itb.ac.id (S.S.); 2Lightweight Structure Research Group, Faculty of Mechanical and Aerospace Engineering, Institut Teknologi Bandung, Jl. Ganesha 10, Bandung 40132, Indonesia; sigit.itb@gmail.com

**Keywords:** silica‒lignin hybrid, oil spill, composites, renewable absorbent material, water treatment

## Abstract

Oil spills in the marine environment are a rising concern due to their adverse impacts on living creatures and the environment. Hence, remediation methods have been used to remove the oil from the contaminated water. A sorbent material is considered the best method for oil spill absorption. However, commonly used commercial sorbents are made from nonrenewable and nonenvironmentally friendly materials. In this research, natural rubber foam (NRF) was used as a sorbent material with the addition of a filler, i.e., silica and a silica–lignin hybrid, to increase its oil sorption capacity and reusability. The silica and silica–lignin hybrid were extracted from rice husk waste by means of the precipitation method. The silica–lignin hybrid-filled NRF exhibited excellent hydrophobicity, with a water contact angle of 133°, and had more stable reusability compared to unfilled NRF and silica-filled NRF. In addition, the optimum oil absorption capacity of silica–lignin hybrid-filled NRF was 1.36 g g^−1^. Overall, the results showed that silica–lignin hybrid-filled NRF has the potential to be developed as a green oil absorbent material and is promising in terms of economic and environmental aspects.

## 1. Introduction

Oil spills in the marine environment have become a crucial issue over the last few decades. They occur due to various reasons such as offshore platforms, fuel leakages from ships, and accidents in pipelines [1]. Additionally, 5.86 million tons of oil are lost globally due to tanker accidents, according to the International Tanker Owners Pollution Federation (ITOPF) [2]. Oil spills are described as the release of hydrocarbon compounds of oil and other chemical traces (sulfur and heavy metals) by accident into the marine environment. The discharge of oil into the seawater can cause harmful impacts on the environment and interfere with the ecological system, which affects the quality of life of living organisms [3,4,5]. Various remediation methods have been used to remove oil in contaminated water, e.g., floating oil booms, skimmers, pumps, in situ burning, dispersants, adsorbents, and bioremediation [1]. The adsorption method is considered the most convenient method for removing oil from polluted water. A sorbent material is categorized as a wettable material with the ability to be wet or non-wet on its solid surface [6,7]. Nevertheless, commonly used materials have demonstrated low separation efficiency and poor separation selectivity [8]. Moreover, these materials are not reusable and cause secondary pollution with toxic gases and land contamination [9]. Therefore, researchers are trying to develop new materials to improve the performance of oil sorbent materials for oil spill absorption.

There are three types of wetting materials developed for oil spill absorption application: oil-removing, water-removing, and smart oil–water separation types. However, the most convenient and common type that has been developed is the oil-removing type [6]. Recently, numerous materials have been investigated for oil spill absorption application, such as graphene foam [10], a melamine sponge [11], a polyurethane sponge [12], a silica aerogel [13], polyvinyl alcohol foam [14], carbon dots/a commercial porous sponge [15], a graphene-based nanocomposite membrane [16], a biomass-based nanofiltration membrane [17], and a PEI/TMSPA/SiO_2_/DTMS fabric [18]. The developed materials exhibited excellent performance and are promising for use in oil spill absorption. However, development is still in progress, despite their excellent properties, since these materials are made of nonsustainable, nonrenewable, and nonenvironmentally friendly materials. One of the promising natural materials that can be developed as a wetting material is natural rubber.

Natural rubber, which consists of cis-1,4-polyisoprene molecules, is a renewable polymer extracted from *Hevea brasiliensis* trees. It is mainly cultivated in tropical regions, such as Southeast Asia and South America [19]. Natural rubber has been widely used in engineering, medical, sports, and household applications, owing to its excellent properties. Natural rubber has good tensile and tear strength, high abrasion resistance, good hydrophobicity [20], and excellent elasticity [19]. Natural rubber has the potential to be investigated for sorbent applications due to its hydrophobic properties, especially for application in oil spill absorption. Chin et al. and Mustapa et al. studied natural rubber foam (NRF), which was a promising material for oil spill application to replace the commercial materials. Unfortunately, the results showed that the oil sorption capacity decreased due to the increase in crosslinking agents [21,22]. To improve its effectiveness, a specific material can be added to the NRF as a filler. The addition of a filler material increases the oil sorption capacity as a result of the surface roughness of the sorbent material that traps oil and other substances [23,24].

Numerous studies on the addition of filler to NRF have been conducted. Songseng et al. reported the fabrication of NRF filled with reduced graphene oxide (rGO), which generated an excellent oil sorption capacity and reusability for oil spill treatments [25]. Zou et al. studied polyethylene aerogel-coated NRF latex that had superhydrophobic and superoleophilic properties [26]. Venkatanarasimhan et al. reported the fabrication of natural rubber with the addition of magnetite nanoparticles for oil spill remediation. The material exhibited good stability and a low water uptake [27]. Riyajan et al. investigated NRF-poly(vinyl alcohol) oil sorption with biodegradable properties [28]. However, these modified oil sorbents still utilized nonrenewable and nonsustainable materials. Therefore, alternative materials are needed in order to substitute the current material used.

Recently, rice husk-based silica has been used as a sorbent in wastewater management applications because of its large surface area and active groups to bind hazardous chemicals [29,30]. The silica in rice husk is widely extracted due to its relatively high content. It is considered an economically cheap and sustainable material compared to commercial silica [31,32]. Nevertheless, according to a previous study, the silica–lignin hybrid material from rice husk has a more significant effect than silica as a sorbent due to its larger surface area and having more active sites [33]. Lignin binds to silica naturally as a matrix via hydroxyl groups. It enhances the physical sorption and acid–base interaction with other substances [34]. Therefore, the ability to absorb oil and other harmful substances is predicted to be better than silica. Furthermore, rice husk is an organic byproduct that is usually burned or wasted in the landfill. It causes harm and damage to the environment. Hence, utilizing the rice husk’s composition for several applications could help prevent such things in the future [35].

To the extent of our knowledge, there is very little information regarding the utilization of the silica–lignin hybrid as a renewable filler in NRF for oil-absorbent applications. This study focuses on the effect of filler on the absorption and reusability of NRF as a sorbent material. In this research, the silica and silica–lignin hybrid were extracted from rice husk through a precipitation process. Three types of sorbent materials were prepared: unfilled NRF, silica-filled NRF, and silica–lignin hybrid-filled NRF. The results show that the silica–lignin hybrid-filled NRF is a promising material for green oil absorption.

## 2. Materials and Methods

### 2.1. Materials

Rice husk (RH) was acquired from a local paddy field in Bandung, West Java, Indonesia. Analytical grade hydrochloric acid (HCl) 37% and sulfuric acid (H_2_SO_4_) 96% were purchased from CV Sopyan Java Cemerlang, Bandung, Indonesia. Sodium hydroxide (NaOH) 98% was purchased from Central Kimia, Bandung, Indonesia. Ribbed smoked sheet 1 (RSS1) was obtained from PT Perkebunan Nusantara VIII, Bandung, Indonesia. Zinc oxide 93–96%, Aflux 42M, and N-cyclohexyl-2-benzothiazolesulfenamide (CBS) 98.50% were purchased from PT Multi Citra Chemindo, Jakarta, Indonesia. The azodicarbonamide blowing agent was provided by PT. Nata Kimindo Pratama, Jakarta, Indonesia. Sulfur 99.9% was obtained from CV Teja Rubber Compounding, Bandung, Indonesia.

### 2.2. Silica and Silica–Lignin Hybrid Extractions

Silica and silica–lignin hybrid materials were extracted using a similar procedure. Silica was extracted from the rice husk ash (RHA) precursor, whereas the silica–lignin hybrid was extracted from the rice husk (RH) precursor. RHA was prepared by a direct combustion process using a furnace at 700 °C for 6 h, whereas RH was prepared without a combustion process. The extraction process was adopted from previous reports with several modifications [34,36,37]. First, the RHA/RH was soaked in 1M of HCl solution with a ratio of 1:16 (*w*/*v*). It was reported that 1M HCl was the most effective agent in removing metallic impurities in rice husk [38]. Then, it was heated and stirred at 95 °C for 90 min. The solution was filtered using Whatman filter paper No. 93, and the residue was washed using tap water to its neutral pH and then dried overnight at room temperature. Afterward, the HCl-treated RHA/RH was mixed with 2M of NaOH solution in a ratio of 1:7 (*w*/*v*). The solution was heated and stirred at 96 °C for 4 h and then filtered using Whatman filter paper No. 93. The filtrate solution was used for the precipitation process. Then, 2M of H_2_SO_4_ solution was added dropwise to the filtrate solution at room temperature until pH 3–4 was reached. Furthermore, the obtained precipitate was left to stand for 6 h at room temperature and filtered with Whatman filter paper No. 93. The final residue was washed using tap water to its neutral pH and dried at 50 °C for 12 h.

### 2.3. Silica and Silica–Lignin Hybrid Characterizations

FTIR spectroscopy characterization was conducted to identify the functional groups of the extracted silica and silica–lignin hybrid. The FTIR Shimadzu Prestige-21 was used at the Laboratory of Analytical Chemistry, Faculty of Mathematics and Natural Sciences, Institut Teknologi Bandung, Bandung, Indonesia. The extracted silica and silica–lignin hybrid samples were mixed with KBr and then formed into a pellet. The samples were inserted into a sample holder and exposed to infrared light in the range of 400–4500 cm^−1^.

The morphology of the extracted silica and silica–lignin hybrid was observed using a Hitachi SU3500 SEM at the Research Center of Nanoscience and Nanotechnology, Institut Teknologi Bandung, Bandung, Indonesia. The samples were prepared in an aluminum stub using adhesive tape and then coated with a layer of gold. The samples were placed into a sample holder and used for imaging. Then, the samples underwent EDX analysis and chemical analysis by exposing them to an X-ray light beam.

The morphology and characteristics of the extracted silica and silica–lignin hybrid particles were also observed using a Hitachi HT770 TEM at the Research Center of Nanoscience and Nanotechnology, Institut Teknologi Bandung, Bandung, Indonesia. The samples were prepared in a suspension form and deposited onto a carbon-coated copper grid. The samples were dried and used for imaging. Then, the samples were used for SAED pattern analysis to determine the crystal structure of the samples.

The specific surface areas of the silica and silica–lignin hybrid were analyzed using the Brunauer–Emmett–Teller (BET) method from the isotherms data via nitrogen adsorption at 77 K. Measurements were conducted using a Quantachrome instrument at the Laboratory of Analytical Instruments, Faculty of Industrial Technology, Institut Teknologi Bandung, Bandung, Indonesia. The samples were vacuum degassed prior to the measurement process. The thermal stability of the extracted silica and silica–lignin hybrid was determined using TG/DTA characterization at the Research Center of Nanoscience and Nanotechnology, ITB, Bandung, Indonesia. The samples were placed into a sample holder with a heating rate of 10 °C/minute from 30 to 1000 °C.

### 2.4. Natural Rubber Compounding

The formulation used in this research is presented in Table 1. The rubber compound was prepared using an XK-160 open mill with a 1:1.35 friction ratio and 13.39 rpm roll speed. The compounding was carried out at room temperature at the Laboratory of Green Polymer, Faculty of Mechanical and Aerospace Engineering, Institut Teknologi Bandung, Bandung, Indonesia. The rubber and additives were prepared by the following procedure reported in Table 2. The samples were labeled according to the filler’s type and part per hundred rubbers (phr).

### 2.5. Natural Rubber Characterizations

The cure characteristics for the rubber compounds were evaluated using the MDR 2000 Alpha at the Research Center for Rubber Technology, Bogor, Indonesia. The testing temperature was 150 °C for 30 min. The torque vs. time curve was recorded to determine the scorch time (ts_2_) and optimum cure time (t_90_). It was calculated from the minimum torque (M_L_) and maximum torque (M_H_).

The morphologies of the NRF before and after swelling in kerosene were characterized using a Hitachi SU3500 SEM at the Research Center of Nanoscience and Nanotechnology, Institut Teknologi Bandung, Bandung, Indonesia. The samples were prepared in an aluminum stub using adhesive tape and coated with a layer of gold. The cross-section area of the NRF was observed. Then, the morphology and average cell size of the NRF were evaluated using ImageJ from the observed SEM images.

The density of the NRF was determined using a geometric method. The relative density (RD) of the NRF was calculated by dividing the density of NRF by the solid natural rubber (0.93 g/cm^3^), as shown in Equation (1). In addition, the volumetric expansion ratio (ER) was calculated as the inverse of the relative density, as shown in Equation (2) [39].
(1)$$\mathrm{Relative}\;\mathrm{Density}\;(\mathrm{RD})=\frac{\rho_{\mathrm{NRF}}}{\rho_{\mathrm{solid}\;\mathrm{natural}\;\mathrm{rubber}}}$$
(2)Expansion ratio (ER)=1ρfoamρsolid

The cell size of the NRF was measured from the observed SEM image and then analyzed using the ImageJ software. Due to the anisotropic shape of the cells, their sizes were measured from two different axes: x (horizontal) and y (vertical). Twenty cells were measured from each sample and noted as average Φ_x_ and Φ_y_.

The contact angle was measured using a digital microscope. Then, 10 µL of water was dropped from a micropipette onto a flat surface of NRF at room temperature. The image was captured and then the contact angle was determined using the Dropsnake method in the ImageJ software.

### 2.6. Oil Sorption Test

An oil absorption test for the NRF was conducted according to ASTM F716-09. The samples were prepared from the vulcanized rubber compound based on t_90_ data. Then, the rubber sheets were cut to a volume of 2 cm^3^ and weighed using an analytical balance. The samples were immersed in 50 mL of kerosene (the viscosity was 1.6 × 10^−3^ Pa s, 25 °C) for 15 min. The NRF was then taken out and hung in the ambient air for 2 min (dripping time), after which it was weighed using an analytical balance. The absorption measurement was recorded as an average from three samples. The absorption capacity was calculated using Equation (3):(3)$$Absorption\;capacity=\left[\frac{W_{2}-W_{1}}{W_{1}}\right]\;g\;g^{-1}$$

To determine its reusability, the absorption and desorption tests were repeated until the NRF was damaged. The samples were squeezed prior to the next cycle of the absorption test. The reusability of the NRF was evaluated using Equation (4) as an average from three samples:(4)$$Reuseability=\left[\frac{\left(W_{2}-W_{1}\right)}{W_{1}}\right]\;g\;g^{-1}\;$$
where *W*_1_ = weight of the unabsorbed material and *W*_2_ = weight of the absorbed material.

## 3. Results and Discussion

### 3.1. Silica and Silica–Lignin Hybrid

#### 3.1.1. Silica and Silica–Lignin Hybrid Extraction

Silica and silica–lignin hybrid were extracted using the precipitation method with different precursor materials. Silica was extracted from the RHA precursor through an incineration process, with the produced yield of around 16.27%. The process was conducted to remove nonsilica contents such as lignocellulose and lignin. The results showed similarities with previous reports [31,40]. Meanwhile, the silica–lignin hybrid was extracted directly from the native RH precursor without the incineration process to keep the lignin embedded within the silica. In this context, the production of the silica–lignin hybrid reduced energy consumption during the preparation process, which was more environmentally friendly compared to the production of silica material. Afterward, the precursor was pretreated with acid in order to remove trace elements such as metal oxides. Afterward, the acid-treated precursor was subjected to an alkaline treatment to separate the silica–lignin hybrid (in liquid form) from the cellulose-rich solid material. According to a previous study [37], the precipitation process was conducted at pH 3–4 due to it obtaining equal silica and lignin composition. Lignin precipitated on the silica’s surface via hydrogen bonding and increased the surface area and the active sites of the hybrid material [41]. The produced yield of the silica–lignin hybrid was around 20.53%, which was higher than the extracted pure silica. This indicated that the extraction process of the native RH precursor produced a higher content than the RHA precursor due to the presence of lignin.

#### 3.1.2. Fourier Transform Infrared (FTIR) Spectra

The FTIR spectra showed that the silica and silica–lignin hybrid have smoother patterns compared to the other spectrum, as presented in Figure 1.

The pattern of the FTIR spectra occurred due to the final composition of the silica and silica–lignin hybrid following the extraction process, which removed lignocellulose and other trace compositions in the RH precursor, as can be seen in Figure 1a. This was confirmed by the decrease in peak intensity at 2926 cm^–1^, 1726 cm^–1^, and 1163 cm^–1^, which represented C–H stretching of the lignocellulose content, C=O stretching of hemicellulose, and C–O–C stretching of cellulose, respectively [42,43]. The silica–lignin hybrid spectra showed a strong absorption peak at 1089.78 cm^–1^, which was assigned to Si–O–Si stretching, and 466.77 cm^–1^ to the bending vibration of Si–O. In addition, the functional group of lignin was assigned at 1641.42 cm^–1^ and 1510.26 cm^–1^. The functional groups of the extracted silica–lignin hybrid are shown in Table 3. The difference between the silica and silica–lignin hybrid spectra is depicted in Figure 1b. The presence of lignin in the silica–lignin hybrid material was presented at the absorption peak of 1510.26 cm^–1^. Meanwhile, the silica spectra only showed the silica-related functional groups. This was also confirmed by measurement using the Klason method (ASTM T222 om-88), which demonstrated that the silica–lignin hybrid consisted of 78.79% silica, 18.93% lignin, and 2.18% impurities.

#### 3.1.3. Scanning Electron Microscope–Energy-Dispersive X-ray (SEM–EDX)

SEM images of the native RH, RH (HCl-pretreated), silica, and silica–lignin hybrid are shown in Figure 2. Native RH exhibited a globular structure on its surface morphology. After HCl pretreatment, it became sharper with no other significant changes, which indicated that the trace elements on the outer surface of the native RH were removed. The SEM image of the silica–lignin hybrid exhibited irregular morphology and was similar to pure silica. However, most of the silica–lignin hybrids possessed smaller particle sizes and rougher surfaces than pure silica. This indicated that lignin molecules were hydrolyzed into smaller fragments embedded onto the surface of silica.

The elemental composition of the silica and silica–lignin hybrid was determined using EDX analysis, as shown in Table 4. Both materials contain silicon, oxygen, and carbon elements. The results show that the 25.83% carbon content from the char component from RHA precursors seems to remain in the extracted silica. This was possibly due to an incomplete combustion process during the preparation of the RHA precursor [38,44]. However, the 9.51% carbon content in the silica–lignin hybrid represented the presence of lignin content as a result of the precipitation process.

#### 3.1.4. Transmission Electron Microscope–Selected Area Electron Diffraction (TEM–SAED)

The morphology of the silica and silica–lignin hybrid was also confirmed by TEM images, as shown in Figure 3. The results show that the silica and silica–lignin hybrid have irregular shapes with tendencies to form agglomerates. The TEM images verified that the lignin consists of small molecules embedded on the surface of silica particles. In addition, according to SAED pattern analysis, the silica and silica–lignin hybrid exhibited polycrystalline and amorphous structures, respectively.

#### 3.1.5. Brunauer–Emmett–Teller (BET) Analysis

The specific surface area of the silica and silica–lignin hybrid is shown in Table 5. According to the BET measurement, the results showed that the surface area of the silica–lignin hybrid was greater than that of the silica material. This confirmed that the presence of lignin had increased the surface area of the silica–lignin hybrid, which resulted in better absorption properties for the absorbent application. The previous study also showed a similar nature, which was reported by Qu et al. [34].

#### 3.1.6. Thermogravimetry–Derivative Thermogravimetry (TG–DTG)

The thermal stability of the native RH, RH (HCl-pretreated), silica, and silica–lignin hybrid was determined using thermogravimetric analysis, as presented in Figure 4. The RH native and RH (HCl-pretreated) showed significant weight loss due to the removal of lignocellulose and lignin contents in the range of 260 to 370 °C and 310 to 385 °C, respectively. Up to 1000 °C, there was still 20% of weight remaining due to the un-decomposed silica content. In addition, the DTG curves of the RH and RH (HCl-pretreated) showed a peak at 335–357 °C, which was similar to a previous report [31]. Conversely, the silica and silica–lignin hybrid have excellent thermal stability at higher temperatures. The extracted silica and silica–lignin hybrid exhibited similar characteristics in the range of 0 to 90 °C with a small weight loss from the materials. However, above 300 °C, the silica–lignin hybrid showed slightly more weight loss than the pure silica as a result of lignin decomposition.

### 3.2. Natural Rubber Foam (NRF)

#### 3.2.1. Cure Characteristics of Natural Rubber Compound

The cure curves and curing characteristics of the natural rubber compounds at 150 °C are presented in Figure 5 and Table 6, respectively. Introducing filler to the natural rubber compounds reduced the maximum torque in both the silica–lignin hybrid and silica samples. The maximum torque of natural rubber compounds decreased along with the increase in phr (part per hundred rubbers) of the filler. This was probably attributed to the decrease in the crosslink density, as indicated in the ∆S’ value. The decrease in the crosslink density was expected due to the acidic nature of both fillers. In addition, the FTIR graph showed that the silica–lignin hybrid and silica contained -OH functional groups. The acidic nature of the functional group tends to delay the vulcanization process [45]. Therefore, a lower crosslink density formed within the natural rubber compounds.

#### 3.2.2. Density and Cellular Structure of NRF

The density and expansion ratio (ER) of the NRF are shown in Table 7. In general, most of the samples showed that the relative density increased as the addition of filler increased. The relative density of the silica–lignin hybrid-filled NRF slightly decreased for the FH-03 and FH-05 samples. This trend was also presented in silica-filled NRF. However, the decrease in density was observed in only the 3 phr sample. The increase in density was expected due to the delayed effect of the silica–lignin hybrid and silica filler during the vulcanization process—the higher the phr of filler added to the NRF, the longer the cure time. Therefore, the decomposition gas of the blowing agent was unable to be trapped inside the rubber. This was also supported by the ER data, which showed that a higher phr of filler decreased the ER value. Thus, only small fractions of cells formed inside the rubber. In this case, only FH-03, FH-05, and FS-05 exhibited an increase in ER value.

The SEM images for all samples are shown in Figure 6. The NRF exhibited closed-cell foam with anisotropic cells and various cell sizes. The addition of fillers, both silica–lignin hybrid and silica, resulted in smaller cell sizes and fewer cells formed as the phr of filler increased. Therefore, more solid natural rubber formed in the samples.

#### 3.2.3. Hydrophobicity of NRF

The hydrophobic and oleophilic characteristics of NRF were studied using water and kerosene droplets on the surface of FS-03 and FH-05 samples (Figure 7). Unfortunately, the contact angle of the kerosene droplet was unable to be determined due to the fact that the kerosene was rapidly spread and absorbed on the surface of the NRF. Nevertheless, the samples showed excellent hydrophobicity.

The water contact angle measurement for the NRF samples is depicted in Figure 8. The water contact angle value of the silica–lignin hybrid-filled NRF increased as the phr increased, which indicated excellent water-repellant properties. In general, the contact angle value of the samples was greater than 100°, and the highest value was 133° for sample FH-09. The unfilled NRF (F-00) sample exhibited a relatively high contact angle at a value of 110°. The value was similar to the result reported by Ratcha et al. [46], of around 117°.

#### 3.2.4. Oil Absorption

The results for the oil absorption capacity using kerosene at room temperature are shown in Figure 9. The addition of filler to the NRF, both the silica–lignin hybrid and silica, led to a higher absorption capacity than unfilled NRF, except for the FS-09 sample. The absorption capacity of the NRF increased as the phr of the filler increased, and then decreased following its maximum absorption capacity. However, the silica-filled and silica–lignin hybrid-filled NRF possessed different maximum absorption capacities. Silica-filled NRF showed maximum absorption capacity at 3 phr by 1.68 g g^−1^, and then decreased to 0.79 g g^−1^ at 9 phr. At that point, the absorption capacity was found to be lower than the unfilled NRF by 0.88 g g^−1^. Meanwhile, the silica–lignin hybrid-filled NRF exhibited maximum absorption capacity at 5 phr by 1.36 g g^−1^. The findings showed a similar result to the previous work reported by Chin et al. [21], which found that the absorption capacity of unfilled NRF in kerosene was in the range of 0.06–0.091 g g^−1^ per minute. In this case, the addition of filler to NRF increased the absorption capacity by 2%.

The oil absorption capacities of NRF depend on the cell size and the surface properties of the material [47]. The larger the cell size formed, the higher the absorption capacity of the NRF. In this case, the silica–lignin hybrid-filled NRF produced larger cell sizes compared to the silica-filled NRF. This is due to the lower density of the silica–lignin hybrid compared to silica. The foam cell can easily form and promotes the formation of large cell sizes. In addition, the presence of lignin in silica enhances the hydrophobicity of the material. Therefore, it also improves the oil absorption capacity of the NRF material.

#### 3.2.5. Reusability of NRF

The absorption capacity at various cycles of the NRF can be seen in Figure 10. According to Figure 10b, the silica–lignin hybrid-filled NRF (FH-05) showed maximum absorption capacity by 1.36 g g^−1^ after the first cycle of exposure to kerosene. For the second cycle, the absorption capacity decreased by 0.33 g g^−1^, which was most likely due to the residual kerosene trapped inside the pores of the NRF. Therefore, the initial weight of the NRF for the next cycle was not accurately weighed. Based on the study, the NRF can be reused 4–5 times before damage. In addition, the reusability of the silica–lignin hybrid-filled NRF was better than the silica-filled NRF.

The SEM images of the NRF after the first absorption test and the last absorption test cycle are presented in Figure 11. After the oil absorption test, the shape of the cells changed into elongated shapes and more small cells appeared at the cross-section of the NRF. In addition, some of the cells collapsed, especially the cells near the surface.

#### 3.2.6. Demonstration of Oil–Water Separation

Oil–water separation by the NRF is shown in Figure 12. The samples were immersed in a mixture of 0.7 mL blue-dyed kerosene and 15 mL tap water in a glass beaker. The samples floated on the surface of the mixture solution and successfully absorbed the oil within 15 min.

Based on the results above, the present work indicates that the silica–lignin hybrid-filled NRF is a promising material to substitute for the current materials used in oil spill absorption applications. In addition, it showed excellent results, especially with respect to economic and environmental factors.

## 4. Conclusions

In this study, a green oil absorbent material was developed as an alternative to the commercial ones. The addition of silica and the silica–lignin hybrid as a filler to the NRF improved the oil absorption capacity of silica-filled NRF and silica–lignin hybrid-filled NRF by 1.68 g g^−1^ and 1.36 g g^−1^ in the first cycle, respectively. Silica–lignin hybrid-filled NRF exhibited excellent hydrophobic properties, with a maximum water contact angle of 133°, as well as stable reusability. The positive results given in this research include (i) the utilization of biomass waste that is harmful to the environment, (ii) insightful ideas on using an all-natural-based material for an oil-absorption application, and (iii) promising results regarding the oil sorption properties of silica–lignin hybrid-filled NRF.

## Figures and Tables

**Figure 1 polymers-14-02930-f001:**
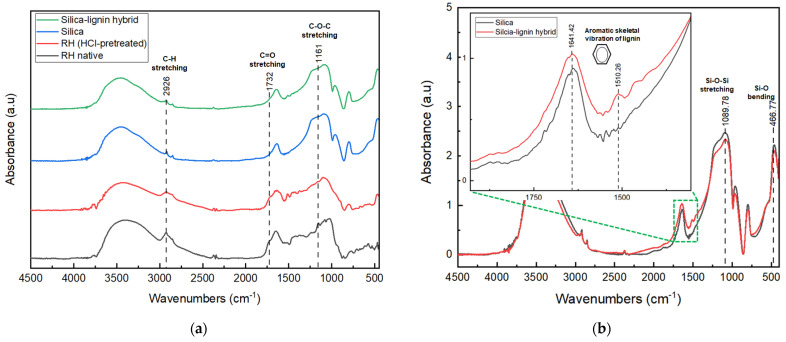
FTIR spectra of (**a**) RH native, RH (HCl-pretreated), silica, and silica–lignin hybrid; (**b**) the comparison peak of silica and silica–lignin hybrid.

**Figure 2 polymers-14-02930-f002:**
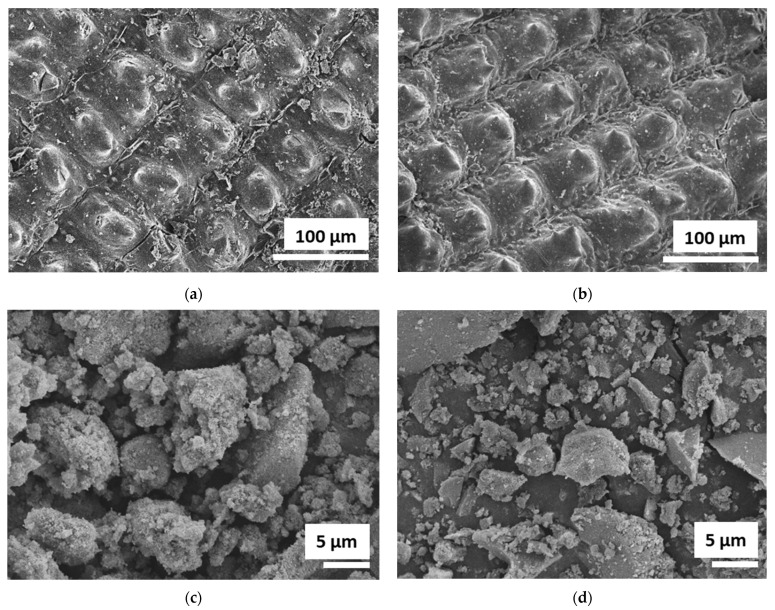
SEM images of (**a**) RH native; (**b**) RH (HCl-pretreated); (**c**) silica; and (**d**) silica–lignin hybrid.

**Figure 3 polymers-14-02930-f003:**
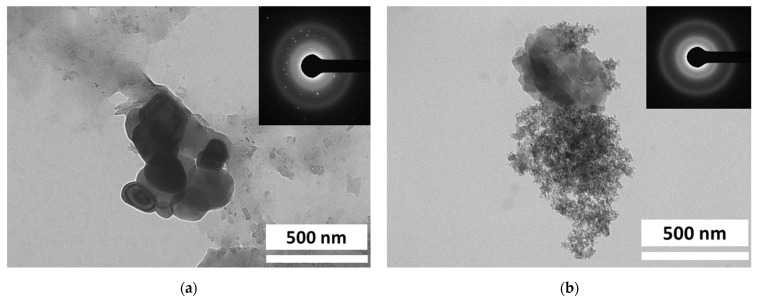
TEM images and SAED pattern of (**a**) silica and (**b**) silica–lignin hybrid.

**Figure 4 polymers-14-02930-f004:**
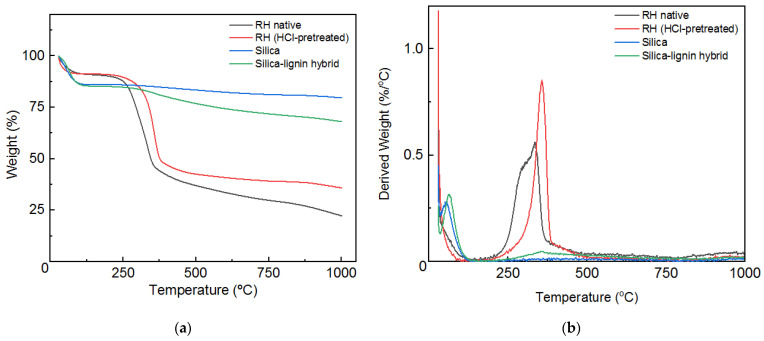
(**a**) Thermogravimetric analysis of RH native, RH (HCl-pretreated), silica, and silica–lignin hybrid and (**b**) DTG analysis of RH native, RH (HCl-pretreated), silica, and silica–lignin hybrid.

**Figure 5 polymers-14-02930-f005:**
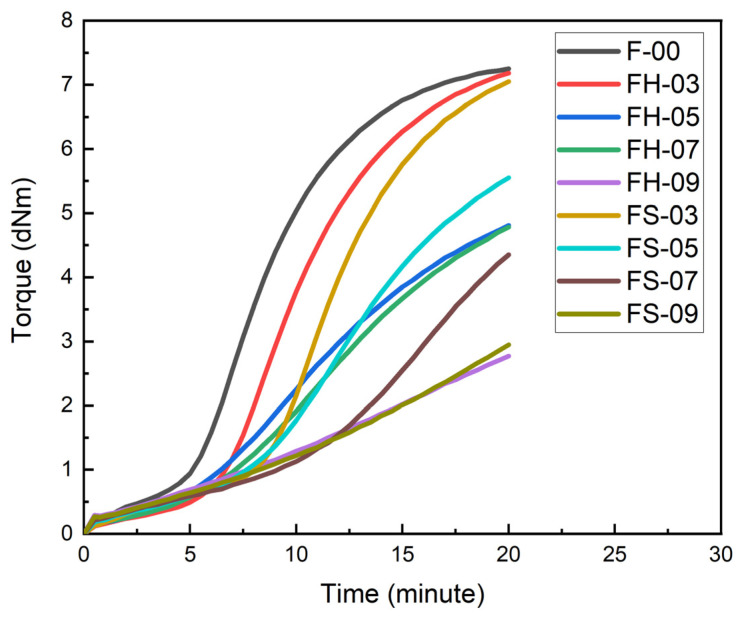
Cure curve of natural rubber compound at various compositions.

**Figure 6 polymers-14-02930-f006:**
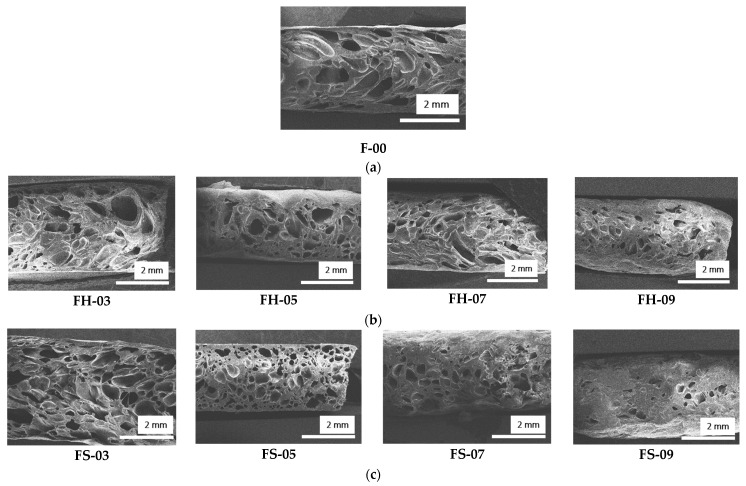
SEM images of (**a**) unfilled NRF; (**b**) silica–lignin hybrid-filled NRF; and (**c**) silica-filled NRF.

**Figure 7 polymers-14-02930-f007:**
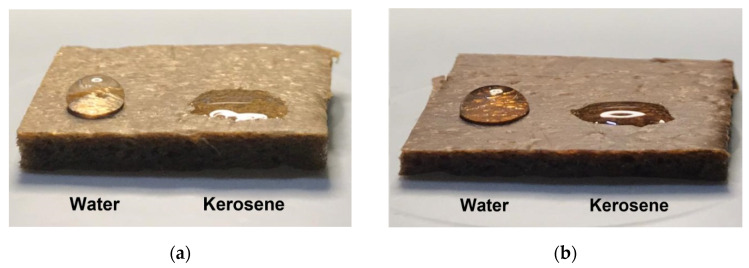
Comparison of the contact angle of the water and kerosene on the NRF surface of (**a**) FS-03 and (**b**) FH-05 samples.

**Figure 8 polymers-14-02930-f008:**
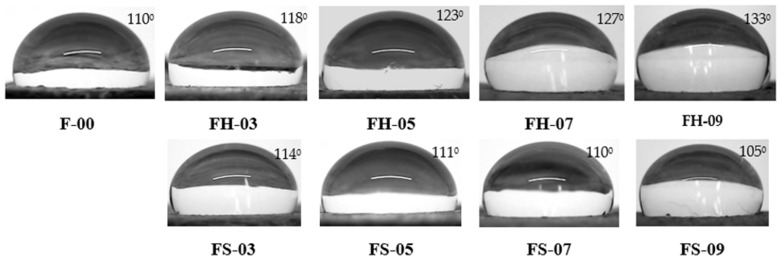
The contact angle of water on the NFR at various compositions.

**Figure 9 polymers-14-02930-f009:**
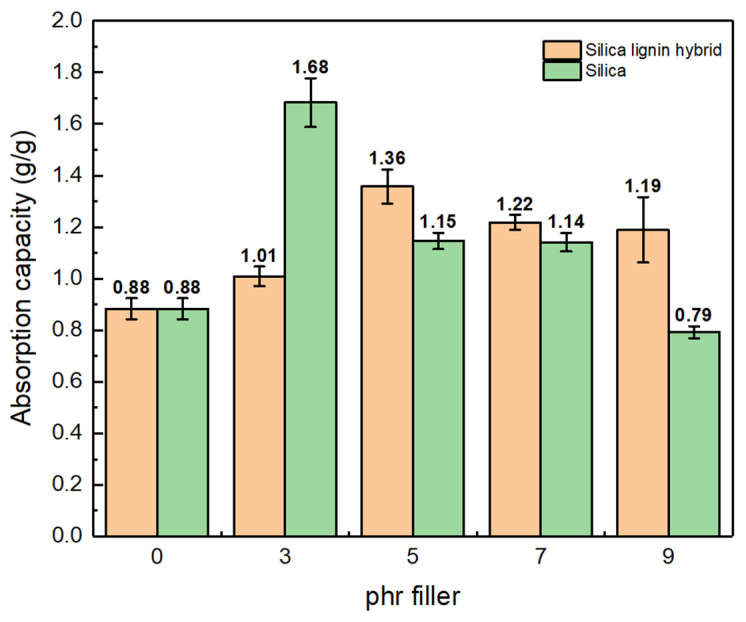
The first cycle of absorption capacity of silica-filled NRF and silica–lignin hybrid-filled NRF at various phr filler contents.

**Figure 10 polymers-14-02930-f010:**
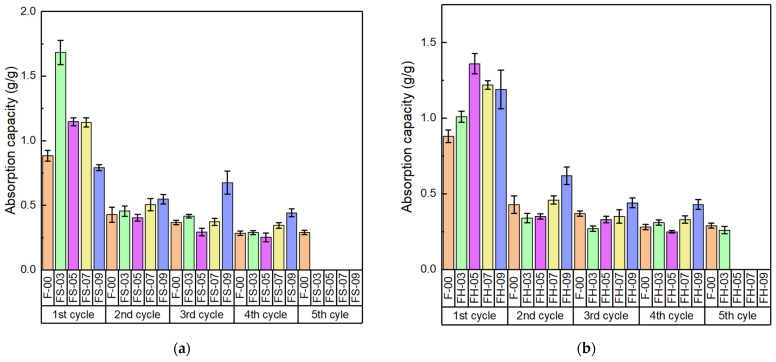
Absorption capacity of (**a**) silica-filled NRF and (**b**) silica–lignin hybrid-filled NRF at various cycles.

**Figure 11 polymers-14-02930-f011:**
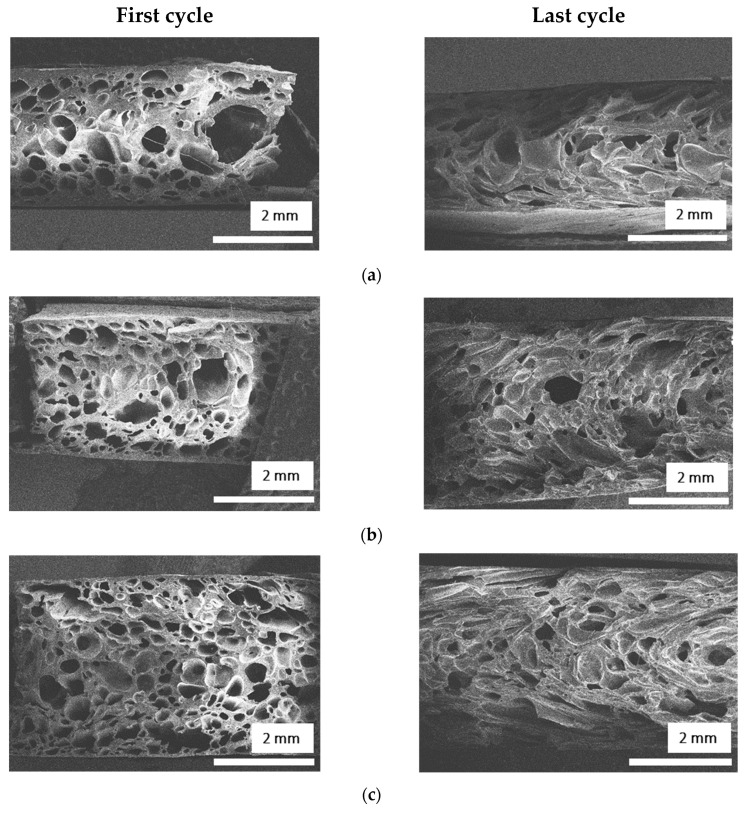
SEM images of NRF after the first absorption test and the last absorption test cycle: (**a**) F-00, (**b**) FH-05, and (**c**) FS-03 samples.

**Figure 12 polymers-14-02930-f012:**
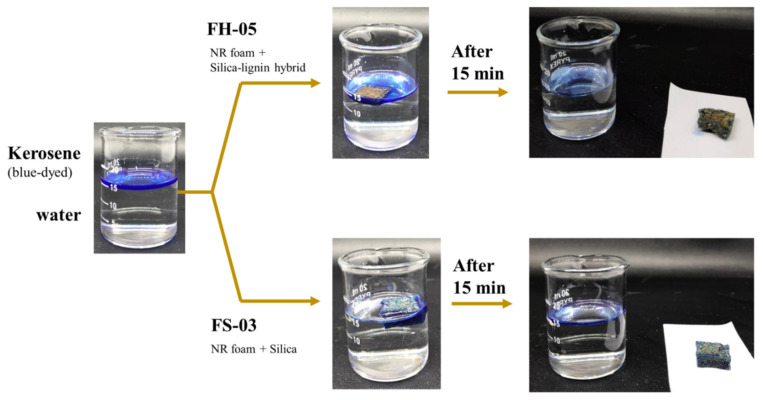
Demonstration of oil–water separation using the optimum silica-filled and silica–lignin hybrid-filled NRF samples.

**Table 1 polymers-14-02930-t001:** Natural rubber foam compounding formulation.

	Formulation (phr)								
	F-00	FH-03	FH-05	FH-07	FH-09	FS-03	FS-05	FS-07	FS-09
RSS1	100	100	100	100	100	100	100	100	100
Aflux 42M	2	2	2	2	2	2	2	2	2
Silica	0	0	0	0	0	3	5	7	9
Silica–lignin hybrid	0	3	5	7	9	0	0	0	0
ZnO	5	5	5	5	5	5	5	5	5
Azodicarbonamide	4	4	4	4	4	4	4	4	4
CBS	0.4	0.4	0.4	0.4	0.4	0.4	0.4	0.4	0.4
Sulfur	2	2	2	2	2	2	2	2	2

**Table 2 polymers-14-02930-t002:** Mixing procedure of natural rubber foam compounding formulation.

Time (min)	Action
2	Rubber mastication
4	Add Aflux 42M
8	Add fillers
2	Add CBS
4	Add ZnO
4	Add Azodicarbonamide
2	Add Sulphur
2	Homogenization

**Table 3 polymers-14-02930-t003:** The functional groups of the extracted silica–lignin hybrid.

Functional Groups	Wavenumber (cm^–1^)	
	Experimental	Reference [41]
O–H stretching (silica–lignin)	3450.65	3390
C–H stretching of methyl and methylene groups	2922.16	2931
	2850.79	2881
Aromatic skeletal vibration of lignin	1641.42	1600
	1510.26	1512
Si–O–Si stretching	1089.78	1087
Si–O stretching vibration	798.53	801
Si–O bending vibration	466.77	568
Si–OH	964.41	948

**Table 4 polymers-14-02930-t004:** Elemental content of silica and silica–lignin hybrid based on EDX analysis.

Sample	Elemental Content (%)		
	Si	O	C
Silica	16.68	57.39	25.83
Silica–lignin hybrid	32.35	57.94	9.51

**Table 5 polymers-14-02930-t005:** Specific surface area of silica and silica–lignin hybrid using BET method analysis.

Sample	Specific Surface Area (m^2^/g)
Silica	245.707
Silica–lignin hybrid	251.526

**Table 6 polymers-14-02930-t006:** Cure characteristics of natural rubber compound for various compositions at 150 °C.

Sample Code	M_L_ (dNm)	M_H_ (dNm)	∆S’ (dNm)	t_s2_ (min)	t_90_ (min)
F-00	0.19	7.34	7.15	6.39	14.2
FH-03	0.12	7.53	7.41	8.09	17.11
FH-05	0.16	5.68	5.52	9.45	22.24
FH-07	0.17	5.78	5.61	10.4	23.04
FH-09	0.27	3.86	3.59	16.36	26.07
FS-03	0.13	7.56	7.43	9.58	18.35
FS-05	0.18	6.6	6.42	10.5	22.28
FS-07	0.22	5.96	5.74	14.06	24.57
FS-09	0.26	4.46	4.20	16.24	26.33

**Table 7 polymers-14-02930-t007:** Density and expansion ratio of NRF at various compositions.

Parameter	F-00	FH-03	FH-05	FH-07	FH-09	FS-03	FS-05	FS-07	FS-09
ρfρs	0.6099	0.5183	0.5177	0.6618	0.6746	0.5699	0.6206	0.6913	0.7786
ER	1.64	1.93	1.93	1.51	1.48	1.75	1.61	1.45	1.28
Φ_x_ (mm)	0.31 ± 0.15	0.36 ± 0.23	0.29 ± 0.18	0.26 ± 0.11	0.24 ± 0.007	0.25 ± 0.13	0.21± 0.12	0.26 ± 0.11	0.23 ± 0.08
Φ_y_ (mm)	0.79 ±0.34	0.72 ±0.27	0.51 ±0.24	0.54 ±0.22	0.38 ±0.09	0.52 ±0.28	0.33 ±0.18	0.44 ±0.17	0.36 ±0.18

## Data Availability

Data presented in this study are available on request from the corresponding author.

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
