# Peer review of "A Silica–Lignin Hybrid Filler in a Natural Rubber Foam Composite as a Green Oil Spill Absorbent"

_polymers, 2022, doi:10.3390/polym14142930_

Round 1

Reviewer 1 Report

This manuscript was focus on the effect of filler on the absorption and reusability of NRF as sorbent material. The authors tried to use renewable material as the filler. However, the explanation for all results in this manuscript is too general. The author should explain in deep details and find the relationship among your results. Several points should also be improved such as

  1. Why the author would like to focus on kerosene?
  2. What is the viscosity of kerosene?
  3. The reusability of materials (FH, FS) is too low….the authors could be reused only 4 times…then it was damaged. It is not accepted for practical application. Why the author did not add some chemical to make high compatibility with NR foam.
  4. Why did the author don’t find the lignin content via Goering and Van Soest method? The FTIR result is not enough. I could not see the difference between silica and silica-lignin.
  5. The explanation for EDX is too short…cannot understand the point of you.
  6. If you know the ratio of silica and lignin, the author can explained more about the mechanism for oil adsorption.
  7. Generally, the presence of filler may prevent the foam formation. However, when adding the filler, the cell size of foam may increase. But in your case showed the different trend, why?
  8. When the authors explained the results, please compare the result between FS and FH.
  9. Could you present the morphology of silica-lignin in NR foam? How about the dispersion and compatibility?
  10. Importantly, why the authors did not present how silica-lignin improve the oil adsorption capacity?

Author Response

We revised the manuscript according to your comments and suggestions, please see the attachment below

Reviewer 2 Report

In this study, the authors reported silica-lignin hybrid as a filler in natural rubber foam compound for more efficient oil spill absorbent. The manuscript is acceptable if authors can be improved the manuscript according to the below comments:

  • Please think of a more specific title.
  • Abstract must be focused on valuable quantitative data for general audiences, which pave the way for persuading the audiences to read the full-text.
  • Keywords should not include the title's contents.
  • The introduction should be targeted at a single main message, and in this format, it is not acceptable for length, scope, and clarity. I strongly recommend for revise of this part according to the below suggestion: Journal of Cleaner Production 347 (2022) 131220 https://doi.org/10.1016/j.jclepro.2022.131220.
  • 1&2&3&4&5&6&7 must be merged as a single high quality figure, representing the characteristics of the samples.
  • 9&10&11must be merged
  • Table 6 should be transformed to an infographics
  • Please note that the captions are desired to be enriched via info, which makes this figure independently understandable instead of relying on the context.
  • Give more up-to-date references with the exclusion of possible self-citations.

Author Response

(The authors gave the same response as above.)

Reviewer 3 Report

Here natural rubber foam was used as a sorbent material. In order to increase oil sorption capacity and reusability, silica and silica-lignin hybrid were added as a filler. Silica and silica-lignin hybrid were extracted from rice husk waste by means of the precipitation method. The work described in the manuscript of sufficient novelty, quality, and potential significance to warrant publication, the provided methods sufficient for an interested reader to reproduce the results, and the conclusions adequately supported by the data presented. So, the publication is recommended after revision:

Comments

  1. Provide the numerical data in the significant findings to highlight the efficiency of the new method.
  2. The authors need to refer to other methods for wastewater treatment in the introduction section; the authors may use: React. Funct. Polym. 65 (2005) 267-275; Sep. Purif. Technol. 43 (2005) 43-48.
  3. Cost of such process needs to be reported and compared with previous related work.
  4. More profound discussions and comparisons with other published works are welcomed.
  5. Procedures followed in the experimental section must be supported by references.
  6. The Manuscript needs thorough revision to improve the text quality and readability of the work.
  7. Add experimental conditions to captions of each figure.
  8. Please check the grammar, uniformity in reference format and spell-check is necessary throughout the manuscript.

Author Response

(The authors gave the same response as above.)

Reviewer 4 Report

  1. The graphical abstract seems to be too complex with may small elements. Simplify it.
  2. The conditions for the preparation were harsh. 700 degC for 6 h, 1 M HCl etc. Can the preparation be optimized and more sustainable?
  3. The authors should consider reporting oil contact angel (e.g. kerosene) and not only water contact angle.
  4. The conclusion section needs to summarize the main research findings in quantitative statements as well.
  5. The novelty of the work is not well communicated. The introduction and the conclusion should stress the novelty of the work in relation to the literature. The pros and cons of the material should also be summarized.
  6. The derivation of the error bars is unclear. Explain in the figure captions how the error bars were derived. Were independently prepared adsorbents used to measure these values? How many of them?
  7. The FTIR spectra presented under Figure 3 need to be annotated.
  8. What is the reason for the low yield in the range of 16-20%? It seems not to be practical. What is the main reason for such a low yield and how can it be improved?
  9. Recent oil spill clean up materials should be briefly mentioned (10.1039/D1NR07111D; 10.1039/D1GC03410C; 10.1016/j.memsci.2020.118007; 10.1039/D1TA08670G).
  10. The purity and/or grade of all materials used in the study should be given under section 2.1 on materials to ensure reproducibility and full interpretation of the research work.
  11. Figure 1 is for a lab book not for a scientific article. It needs to be deleted. Two datapoints do not qualify for a figure. The yield values can simply be mentioned in a sentence. Delete Figure 2.
  12. Overall the adsorption capacity significantly declined during the reuse of the material. What is the explanation for that and how can it be improved? The authors should consider the use of ultrasound to regenerate the material, which a green way and can result in better cycling performance.

Author Response

(The authors gave the same response as above.)

Round 2

Reviewer 2 Report

The authors did not make a thorough during the revisions, and they bragged via highlighting extensive parts while these parts were included minor corrections. Please employ the comments of all reviewers one by one without compromising. Error bars are questionable in some of the bar charts. Fig 2 SEM images of (c) silica; and (d) silica-lignin hybrid must be retaken plus TEM and SAED. BET of silica and silica-lignin hybrid must be added. Fig 11 is very poor. MDPI rule should be refered via link. RH unsusal curve (Figure 3b) must verified by the literature.

Reviewer 4 Report

Previous comment 1: It can be still futher simplifed, it has too many small elements.

Previous comment 2: The answer is acceptable but such argument should be briefly provided in the mauscript as well.

Previous comment 4: Now 2 percentage values are given. Percentage values should not be used as they are meaningless. Adsorption should be reported as g/g in the conclusion and throughout the manuscript.

Previous comment 7: Only 1510 cm-1 is highlighted still. Annotation requires all the major peaks to be noted (value and meaning).

Previous comment 9: Two examples are still missing.

Round 3

Reviewer 2 Report

changes were not desirable

Author Response

Dear reviewer,

We would like to send our gratitude to you for taking your precious time to review our manuscript. The manuscript has undergone English language editing, as seen in the latest revised manuscript. We would like to thank you again for reviewing our manuscript.

Best regards,

Mardiyati

Reviewer 4 Report

The comments have been addressed.

Author Response

(The authors gave the same response as above.)
